# Application of a Newly Developed Chitosan/Oleic Acid Edible Coating for Extending Shelf-Life of Fresh Pork

**DOI:** 10.3390/foods11131978

**Published:** 2022-07-04

**Authors:** Van-Ba Hoa, Dong-Heon Song, Kuk-Hwan Seol, Sun-Moon Kang, Hyun-Wook Kim, Jin-Hyoung Kim, Sung-Sil Moon, Soo-Hyun Cho

**Affiliations:** 1Animal Products Utilization Division, National Institute of Animal Science, RDA, Wanju 55365, Korea; b1983@rda.go.kr (V.-B.H.); sdh8507@rda.go.kr (D.-H.S.); seolkh@rda.go.kr (K.-H.S.); smkang77@rda.go.kr (S.-M.K.); woogi78@rda.go.kr (H.-W.K.); jhkim702@rda.go.kr (J.-H.K.); 2Sunjin Meat Research Center, Ansung 17532, Korea; ssmun@sj.co.kr

**Keywords:** pork, chitosan, oleic acid, coating, shelf-life, discoloration

## Abstract

This study aimed at evaluating the applicability of a newly-developed chitosan/oleic acid edible coating for extending the shelf-life of fresh pork under aerobic-packaging conditions. Various coating formulations were used: 2% chitosan alone (CHI), 0.5% (*v*/*v*) oleic acid in 2% chitosan (CHI/0.5%OA) and 1% (*v*/*v*) oleic acid in 2% chitosan (CHI/1%OA) were prepared. For coating, fresh pork slices were fully immersed in the coating solutions for 30 s and dried naturally at 4 °C for 30 min. The coated samples were placed on trays, over-wrapped with plastic film, stored at 4 °C for 21 days, and were analyzed for shelf-life stability. Samples without coating were used as control. It was found that the aerobic bacteria and *Pseudomonas* spp. counts, and total volatile basic nitrogen (TVBN) content were almost two to three times lower in the CHI/OA-coated samples compared to the control after 21 days of storage (*p* < 0.05). The CHI/OA coating combination completely inhibited growth of *E. coli*, and protected the meat from discoloration after 21 days of storage. In particular, the addition of OA increased the concentration of volatiles associated with pleasant aromas. This study provides an application potential of chitosan/oleic acid edible coating in preservation of fresh pork to prolong the shelf-life and improve safety.

## 1. Introduction

Meat is a perishable food type by nature involving the actions of microorganisms and endogenous enzymes that cause physicochemical composition changes [1]. During slaughter, fabrication and processing, raw meat is easily contaminated with pathogenic bacteria that may result in foodborne diseases and public health concerns [2]. Among meat types, pork is the most consumed in many countries worldwide [3]. Due to its rich nutrients, the pork is a desirable environment for the bacterial growth that leads to discoloration, its off-taste and texture change, etc. [4]. The predominant bacterial species associated with storage of raw pork are *Pseudomonas* spp. and *Enterococcus* spp. etc. [5]. The actions of proteases released from the spoilage bacteria cause the protein and nitrogen-containing compound degradations which result in the accumulation of organic amines and volatile basic nitrogen [1]. Many of these compounds cause negative effects on human health, discoloration and off-flavor of meat and meat products [6,7]. Moreover, lipid oxidation is also known to be one of spoilage mechanisms causing quality loss, change in flavor and change in the color of meat [8]. To reduce these spoilage mechanisms, some synthetic antimicrobials and antioxidants have been used in the meat industry [9]. However, the utilization of synthetic antimicrobials and antioxidants have been suspected to cause health hazards [10]. In this context, the application of natural antimicrobial and antioxidant—based preservation techniques are needed to retain freshness and quality of meat as well as reduce the negative effects on consumer health.

In recent years, there has been an increasing interest in the application of non-thermal deactivation techniques such as edible coatings for the preservation of fresh meat [11,12]. The edible coatings are thin layers of edible materials applied on food surface, that act as a barrier against microbial growth and physicochemical deteriorations [13]. Amongst, chitosan, used as a food additive in many countries [14], is a film-forming biodegradable polymer obtained from crustacean shells [15]. Chitosan has a potent antimicrobial activity against almost all bacteria species and fungi [15,16]. Because of its low cost and highly protective effect, the chitosan-based coatings have recently been used as packaging material for the preservation of meat. Kanatt et al. [17], Duran and Kahve [18], and Alirezalu et al. [19] reported that coating with 2% chitosan improved the microbiological quality of chicken, beef and pork during storage. Researchers have found an enhanced efficiency of chitosan coatings on the preservation of meat during storage when it is combined with other antimicrobial and antioxidant compounds (e.g., ε-polylysine and gallic acid) [19,20].

Fatty acids (FA) are naturally found in the animal and plant cells. Many reports have demonstrated that FAs all exhibit a broad-spectrum inhibitory activity against bacteria and fungi, etc. [21,22,23]. Out of FAs, oleic acid (a monounsaturated FA, C18:1n-9) possesses a potent activity against both spoilage and pathogenic bacteria (e.g., Staphylococcus aureus, *E. coli*, *Helicobacter,* Clostridium and Salmonella species etc.) [22,23,24]. More importantly, reports have shown that a higher content of oleic acid is associated with better flavor characteristics of meat [25], since it contributes significantly to the development of cooked meat aromas [26]. Until now, oleic acid has also been used in chitosan-based coatings for reducing quality loss and prolonging the shelf-life of fruits [13,27]; however, no attention has been paid to the investigation of its application in the preservation of fresh meat. Thus, the main objective of this study was to assess the effects of the chitosan/oleic acid coating combination on the shelf-life and aroma compounds of pork during storage.

## 2. Materials and Methods

### 2.1. Preparation of Coating Solutions and Application

The chitosan solutions were prepared by adding 2% chitosan (medium molecular weight, Sigma-Aldrich, St. Louis, MO, USA) into 1% (*v*/*v*) acetic acid solution (Sigma-Aldrich, St. Louis, MO, USA), which was stirred for 24 h at room temperature. Thereafter, different formulations of coating solutions with chitosan/or oleic acid (OA) (Sigma-Aldrich, St. Louis, MO, USA): 2% chitosan solution alone (CHI), 0.5% (*v*/*v*) OA in 2% chitosan solution (CHI/0.5%OA) and 1% (*v*/*v*) OA in 2% chitosan solution (CHI/1%OA) were prepared. The concentration of OA set in the present study was based on its antimicrobial activity reported in previous studies [28]. The coating mixtures were then homogenized at 11,000 rpm for five min using a homogenizer (Polytron MR-2100, Bern, Switzerland), and then their pH was adjusted to 5.8 with sodium bicarbonate (Sigma-Aldrich, St. Louis, MO, USA). To ensure film forming ability, the coating solutions were analyzed for their morphological properties. For this, approximately 0.1 mL of each coating solution were placed in silicone, dried at 4 °C for 24 h and coated with platinum for analysis by scanning electron microscopy (SEM) (Supra 40 VP Instrument, Zeiss Co., Oberkochen, Germany), proving the ability of both to form films (Figure 1A,B).

Whole shoulder butt cuts (*n* = 20, approximately 80 kg) from male Woori black pigs were purchased at 24 h post-mortem from a local slaughterhouse (Jeonju, Korea). After removing about 3.0 cm from each end of each cut, they were prepared into 2-cm thick slices (*n* = 160, approximately 400 g/slice). The slices were fully immersed in the coating solutions for 30 s and dried naturally at 4 °C for 30 min. To determine the preservative efficacy (antioxidant and antimicrobial properties) of treatments and to accelerate the spoilage mechanisms, all of the coated meat samples were subjected to aerobic packaging. Particularly, after placing them on foam trays (two steaks/tray), they were wrapped with polyvinyl chloride film. The non-coated meat samples were used as control (CON). Thereafter, the samples were stored at 4 °C for one (the day after coating), seven, 14 and 21 days. For each storage period, forty samples (*n* = 10/treatment or five trays/treatment) were randomly taken and used for analysis of shelf-life and quality properties.

### 2.2. Shelf-Life Measurements

#### 2.2.1. Microbiology

Approximately 10 g samples taken aseptically from different areas of each slice were homogenized with 90 mL saline for 1 min using a Stomacher (Bag mixer 400W, Saint-Nom, France). A progressive dilution of ten times with saline solution was then performed. The total aerobic plate count (APC) was determined on a Petrifilm Aerobic Count Plate (3M Health Care; Saint Paul, MN, USA) after incubating at 37 °C for 48 h. *Pseudomonas* spp. were enumerated on Pseudomonas Agar Base (Oxoid Ltd., Hants, UK) containing selective agar supplement (RS0103) after incubated at 30 °C for 48 h [29]. Total E. coli was determined using Petrifilm *E. coli* Plate Count (3M Health Care; Saint Paul, MN, USA) after 48 h of incubation at 37 °C. The results were calculated and expressed as logarithms of number of colony forming units per gram of meat (log cfu/g).

#### 2.2.2. pH Measurement

To determine the pH of the samples, a pH meter (model: pH*K 21, NWK-Technology GmbH, Kaufering, Germany) was used, which was previously calibrated with pH 4 and seven buffer solutions. Each the samples was measured by inserting the probe deeply into three different locations of the muscle tissues.

#### 2.2.3. Total Volatile Basic Nitrogen (TVBN)

To determine the degradation degree of proteins and nitrogen-containing compounds in the meat during storage, the TVBN content was measured using the method as described in our previous study [30]. Briefly, after grinding, the samples (5.0 g each) were homogenized with distilled water (45 mL) at 11,000× *g* rpm for 30 s. The homogenates were filtered with Whatman filter paper, and 1.0 mL of sample was applied onto the outer space of the Conway tool. Next, 1 mL of 0.01 N boric acid and 100 µL of Conway reagent were applied onto the inner space. Finally, 1 mL of 50% (*w*/*v*) potassium carbonate solution was added into the outer space. After sealing, the Conway tool was incubated at 37 °C for 2 h. For determination of TVBN, various volumes of 0.02 N sulfuric acid solution were added onto the inner space until the color changed to violet. The TVBN content was calculated and expressed as mg TVBN/100 g meat.

#### 2.2.4. Thiobarbituric Acid Reactive Substances (TBARS)

The TBARS assay was carried out using the method of Buege and Aust [31] to measure the extent of lipid oxidation in the samples during storage as described by Hoa et al. [30]. Results were reported as mg malondialdehyde/kg meat (MDA/kg). Each sample was measured in triplicate.

#### 2.2.5. Meat Color

Color was determined by a colorimeter (Konica Minolta Chroma Meter CR-400, Tokyo, Japan) fitted with a D65*C illuminant and a second observer (Minolta Camera, Osaka, Japan). After removing the plastic film from the trays, the CIE L* (luminosity) and a* (redness) values were measured directly at five different points on the slice surface without removing the coating. Throughout storage, the reduction in the a* value (redness) of the slices was also calculated using the measured a* values expressed as a percentage of the initial a* values (day one).

### 2.3. Fatty Acid Composition

To determine whether the incorporation of oleic acid affects the fatty acid profiles of coated meat, the fatty acid profiles were analyzed. The meat samples (10 g each) with edible coatings at one day storage were weighed and extracted for lipids by chloroform/methanol (2:1) as described in our previous study [32]. Briefly, the extracted lipid fractions were initially methylated with Na_2_SO_4_ which was followed with a mixture of tricosanoic acid/0.5 N NaOH solution (1:1 ratio). The methylation procedure was carried out at 55 °C for about 20 min. The methylated fatty acids were separated on a column (30 m × 0.25 mm × 0.25 µm; Supelco) connected with a gas chromatography/flame ionization detector (Varian Technologies) under conditions as described in the above-mentioned reference [32]. Individual fatty acid was expressed as a relative percentage of total fatty acids.

### 2.4. Aroma Volatiles

To examine whether the coatings affect the aroma characteristics of the meat, volatile aroma compounds were analyzed using the method as described in our previous study [30]. Briefly, after cooking on an open tin-coated grill at around 180 °C for 2 min, 2.0 g of each sample was weighed, put into 20-mL headspace vial and capped with silicone septum. The extraction of volatile compounds was carried out at 60 °C for 50 min using a 75 μm carboxen–polydimethylsiloxane fiber (Supelco) connected to a SPME auto-sampler (model: PAL RSI 85). The extracted volatiles were then separated using gas chromatography (model: 8890 GC system) with mass spectrophotometry (5977B MSD, Agilent Technologies). The GC/MS conditions, identification and quantification methods of volatile compounds were the same as those used in the aforementioned reference [30].

### 2.5. Statistical Analysis

Data was analyzed using a two-way ANOVA of Statistical Analysis System (SAS Institute, Cary, NC, USA, 2018). A random block design was used to analyze the obtained results, considering a mixed linear model as a random effect, and coating treatment and storage period as fixed effects. For each storage period, forty samples (10 samples/treatment) were evaluated. Duncan’s Multiple Range Test was used to compare the means, and differences among the mean values were set at *p* < 0.05.

## 3. Results and Discussion

### 3.1. Fatty Acid Profiles

Our results (Table 1) showed that the level of oleic acid was significantly higher in the samples coated with CHI/1% OA compared to the other groups (e.g., CON and CHI) (*p* < 0.05). This could be due to the coating materials (CHI/1% OA) present in the meat sample. As a result, the total monounsaturated fatty acids (MUFA) content and MUFA/saturated fatty acids ratio were higher in the CHI/1%OA group compared to the other remaining groups (*p* < 0.05). Fatty acid profiles not only indicate the nutritional value but also remarkably influence the development of meat flavor during cooking [33]. The MUFAs (e.g., oleic acid) in meat produce various volatiles which contribute to desirable aromas of cooked meat [34]. Thus, the coating with CHI containing OA could improve the nutritional value and is expected to enhance the aromas of coated meat.

### 3.2. Effects on Microbial Quality

Total aerobic plate count (APC, overall bacterial populations) and *Escherichia coli* (*E. coli*) are usually considered as important indicators reflecting the sanitary and safety status; their high counts indicate poor sanitation control and the short shelf-life of raw meat products [35,36]. Aerobic packaging accelerates the spoilage process of meat because it favors the growth of *Pseudomonas* spp. [37]. Both APC and *Pseudomonas* spp. have been found to be major candidates causing the spoilage of meat during refrigerated storage under aerobic packaging conditions [1]. Results (Table 2) showed that at the first day of storage, the APC, *E. coli* and *Pseudomonas* spp. counts were not different between the control and coated samples (*p* > 0.05). After seven and 14 days of storage, the APC, *E. coli* and *Pseudomonas* spp. were lower in the coated samples in comparison with the control (*p* < 0.05). After 21 days of storage, the APC (7.61 log cfu/g), *E. coli* (2.61 log cfu/g) and *Pseudomonas* spp. (5.58 log cfu/g) were almost two to three times higher in the control compared to the coated samples (*p* < 0.05). Compared to the coating with chitosan alone, the addition of oleic acid produced a higher inhibitory effect against the microbial growth. Particularly, after seven, 14 and 21 days of storage, all of the CHI/OA-coated samples exhibited a lower APC, *E. coli* and *Pseudomonas* spp. levels than the CHI-coated samples (*p* < 0.05). The incorporation of 0.5 or 1% oleic acid completely inhibited the growth of *E. coli* after 21 days of storage. We also found that the higher the dose of oleic acid added, the higher the inhibitory efficiency against the APC and *Pseudomonas* spp. growth.

With prolonged storage time, the APC (2.43, 4.05, 5.71 and 7.61 log cfu/g at 1, 7, 14 and 21 days of storage, respectively) and *Pseudomonas* spp. (1.97, 3.96, 4.57 and 5.58 log cfu/g at 1, 7, 14 and 21 days of storage, respectively) in the CON continuously increased (*p <* 0.05). Whereas the coated samples showed a slower rate of APC and *Pseudomonas* spp. growth; the number of APC and *Pseudomonas* spp. only increased during the first seven days and decreased thereafter to a level that was close to the initial number (1st day 1) after 21 days of storage. Furthermore, although most *E. coli* are generally non-pathogenic and ubiquitous to all animals, some serotypes (e.g., *E. coli* O157 etc.) can cause foodborne disease [38]. Our result showed that at day one of storage, the entire sample presented a relatively low *E. coli* count (1.5 log cfu/g), and it increased in the CON after seven days and remained unchanged thereafter. In the CHI-coated samples, the *E. coli* was still found at a relatively low level (1.36 log cfu/g), but it was absent in the CHI/OA-coated samples at the end of the storage period (21st day). This signifies that the addition of oleic acid had a higher inhibitory effect against *E. coli* growth in the pork during storage. Similar to the current results, some studies showed that chitosan coating inhibits spoilage bacteria growth in pork, beef and chicken under different packaging conditions [18,20,30,39]. The mechanism behind this phenomenon may be due to the antimicrobial activities of chitosan and oleic acid. The chitosan’s antimicrobial activity is due to its interaction with negatively charged microbial cell membranes which leads to the leakage of intracellular contents, and subsequently cell death [40]. While the antimicrobial activity of oleic acid has been attributed to its interaction with bacterial cell membranes leading to increased cell membrane permeability and disruption of the electron transport chain and inhibition of enzyme activity etc. in the bacterial cells [21,22,23]. Furthermore, the differences observed between the CHI and CHI/OA treatments in terms of their influence on microbial parameters may be due to the synergistic antimicrobial activity by both chitosan and oleic acid. Similarly, Fang et al. [20] and Hoa et al. [30] also showed that coating with chitosan containing natural antimicrobials and antioxidants enhances the antimicrobial activity against spoilage bacteria growth in meat during cold storage when compared to chitosan coating alone. According to safety guidelines in terms of microbiological quality for most meat types (pork, beef and chicken etc.) in different countries (Korea, USA, Australia and European Union etc.), the maximum limit of APC and *E. Coli* should not exceed 5–7 log cfu/g and 2–4 log cfu/g, respectively [41]. Thus, after 21 days of storage, all of the coated samples had the APC and *E. Coli* levels that were lower than the recommended limits, whereas the non-coated samples showed the APC that was higher than the recommended limit after 14 days of storage.

### 3.3. Effects on TBARS and TVBN Contents

It is well recognized that the lipid oxidation causes the off-flavors and discoloration of meat [8]. As shown in Table 3, no differences in the TBARS values occurred between the non- and coated- samples at the beginning (first day) (*p* > 0.05). From the seventh to the 21st day, all of the coated samples displayed a lower TBARS value compared to the control (*p*
*<* 0.05). After 21 days of storage, the TBARS value was in the following order: CON (1.03 mg/kg) *>* CHI (0.49 mg/kg) *>* CHI/0.5%OA (0.46 mg/kg) *>* CHI/1%OA (0.45 mg/kg). Thus, the control increased TBARS content by 0.7 mg, whereas the coated samples only increased by 0.12 to 0.16 mg after 21 days of storage. This finding agrees with that of Fang et al. [20] and Hoa et al. [30], who also reported a lower TBARS value in chitosan-coated meat (pork and beef) under vacuum or aerobic packaging conditions during storage. Georgantelis et al. [42] reported that when the TBARS value is above 0.6 mg/kg, the off-flavors in meat can be detected by consumers. In this study, the TBARS values in all of the coated samples were below this limit, except the control after 21 days. On the other hand, from the 14th to 21st day, the TBARS value was lower in the CHI/1% OA compared to the CHI, indicating that the addition of 1% OA exhibited a greater protective effect against the lipid oxidation during storage. The mechanism underlying this phenomenon may be due to the chitosan/oleic acid mixture that forms a barrier to cover the meat and thus reduces the oxygen permeability and the lipid oxidation [38].

TVBN content is considered to be one of the most important indices indicating the freshness of meat [1]. As shown in Table 3, the TVBN values (7.36–7.86 mg/100 g) were the same in both non-and coated-samples at the beginning (day one) (*p* > 0.05). From the seventh day to the end of storage, the coating showed its effects against the TVBN formation compared to the non-coating. Compared to the CHI alone, the CHI/OA combination also exhibited higher inhibitory effects against TVBN formation, and this effect was further enhanced as the oleic acid level increased to 1% after 21 days of storage. At the end of storage (day 21), the TVBN value was in the following order: CON (56.01 mg/100 g) *>* CHI (27.53 mg/100 g) *>* CHI/0.5% OA (19.76 mg/100 g) *>* CHI/1% OA (17.41 mg/100 g). Thus, the control showed the fastest rate of TVBN increase (by 48.52 mg), followed by CHI (by 20.11 mg), CHI/0.5%OA (by 12.40 mg) and CHI/1% OA (by 10.01 mg) after 21 days of storage. The VBN are organic amines that form in the degradation of proteins and others (e.g., nucleic acids) by spoilage bacteria, and they cause meat’s discoloration and off-flavor as well as constituting a health hazard [1]. Therefore, the results indicating the lower TVBN values in the coated samples during storage could be attributed to the antimicrobial properties of both chitosan and oleic acid as mentioned above [21,23,40]. According to the guidelines for freshness criteria by Korea (upper limit of 20 mg/100 g) [1], the CON only remained fresh for seven days, while the CHI-coated samples remained fresh for a longer time (14 days), and those coated with CHI/1% OA could remain fresh for up to 21 days under aerobic packaging condition. Similar to the current finding, a lower TVBN content was found in chitosan-coated meat compared to control (non-coating) during storage as reported by Cheng et al. [43] and Hoa et al. [30].

### 3.4. Effects on Color and pH

Color is the most important quality trait indicating the freshness of meat [41]. The effects of coating on the color traits of pork during storage are summarized in Table 4.

Regarding L* (lightness), no differences in its values occurred between the CON and coated-samples within the first seven days of storage (*p* > 0.05). From the 14th to the 21st day, the CON showed a lower L* value (48.29–50.29) compared to the coated samples (51.08–53.31) (*p <* 0.05). The CHI/OA maintained a stable L* value unchanged throughout the storage period (21 days) (*p* > 0.05). Contrastingly, the CON continuously decreased the L* values from day 7 to 21 of storage. Compared to the CHI/OA coating combination, the CHI alone showed an increase in L* value with increased storage time, which is similar to the findings of Hoa et al. [30] with regard to beef. These results indicate that the CHI/OA coating combination could retain the lightness of fresh pork for up to 21 days of storage (as shown in Figure 1).

A bright red color is known as the positive trait for the freshness of meat, and the degree of redness (a*) of meat is heavily depended on the myoglobin redox forms [44]. No differences in the a* values occurred between the CON and coated-samples at one day of storage (*p* > 0.05). From the 7th to the 21st day, all of the samples coated with chitosan/or oleic acid showed a higher a* value compared to the CON (*p <* 0.05). Interestingly, compared to the CHI coating alone, the addition of 1% oleic acid showed a higher a* value (16.69–17.16) on all the examining days (*p <* 0.05). During storage, the CON showed the quickest rate of a* values reduction; corresponding to a loss by 12.87, 20.14 and 38.59%, after seven, 14 and 21 days of storage, respectively (*p <* 0.05) (Table 4). CHI maintained a relatively stable a* value during the first 14 days and tended to decrease thereafter (corresponding to a loss by 3.76, 10.04 and 15.61%, after 7, 14 and 21 days of storage, respectively), whereas, the CHI/1%OA could maintain a stable a* value unchanged throughout the storage period (21 days) (*p* > 0.05). After 21 days of storage, the loss of a* values was in the following order: CON (38.59%) *>* CHI (15.61%) *>* CHI/0.5%OA (7.90%) *>* CHI/1% OA (5.63%) (*p*
*<* 0.05). Consistent with our results, other studies have also showed a slower rate of redness loss in chitosan-coated pork [20] and beef [30] compared to control (non-coating) during storage. The mechanisms behind these obtained results may be due to: (i) under the aerobic packaging, the coatings could retard the oxygen permeation into the meat, which stabilized the color by delaying the oxidation of oxymoglobin to metmyoglobin [45]; (ii), the inhibitory effects against the growth of spoilage bacteria by the coatings because the discoloration of meat during storage usually results from the oxidative processes of protein and lipids and growth of spoilage bacteria [1,41]. As mentioned above, the CHI/0.5% OA and CHI/1% OA showed the lowest spoilage bacteria count (Table 2), TBARS and TVBN levels (Table 3) compared to the CON and CHI. Overall, the chitosan/1% oleic acid coating combination exhibited a superior protective effect against the discoloration of pork buttock during storage as compared to the chitosan coating alone (as seen in Figure 1).

The pH significantly affects the color and other quality attributes of meat [1]. During the first seven days of storage, no differences in pH values occurred between the control and coated samples (*p <* 0.05). After 14 to 21 days, the samples coated with CHI/OA showed lower pH values compared to the CON and those coated with chitosan alone (*p <* 0.05). The CON and CHI increased in their pH (*p*
*<* 0.05), whereas the CHI/0.5%OA and CHI/01%OA retained the same pH throughout the storage period (*p* > 0.05). It is well recognized that the basic chemical compounds (e.g., TVBN) formed in the degradative processes by endogenous and microbial enzymes is responsible for the pH increase in meat during storage [1].

### 3.5. Effects on Aroma Volatiles

Aroma volatiles generated during cooking significantly contribute to the development of cooked meat flavors [33]. Thermal oxidation of fatty acids during cooking/heating also play an important role in the cooked meat flavors development [34]. A small change in the fatty acid composition of meat may result in alteration to the aroma volatiles during cooking [46]. The effects of CHI/OA coating on the aroma volatiles of cooked meat evaluated at both storage periods (the first and 21st days of storage) are shown in Table 5. Aldehyde was the most predominant class of volatiles, with 23 compounds identified. Out of them all, octanal, nonanal and decanal are the major compounds of oleic acid during the cooking of meat [26]. These aldehydes are very important for cooked meat flavors, as they are characterized by pleasant fatty–sweet–oily aromas [47,48]. Results showed that the concentrations of octanal and nonanal were higher in the CHI/0.5% OA and CHI/1% OA than those in the CON and CHI on both day one and 21 of storage (*p <* 0.05). Some aldehydes such as butanal and 2-ethylhexenal were found in the CON, whereas they were absent in the coated samples after 21 days of storage. These two aldehydes have been reported to be produced by spoilage bacteria during the storage of meat [46]. The coating affected the levels of five alcohols such as 1-pentanol, 1-hexanol, 1-heptanol, 1-octen-3-ol and 1-octanol. Of these, 1-octanol is known as the oleic acid-derived product and the other remaining alcohols are the linoleic acid-derived products during cooking [26]. The alcohols are important for the cooked meat flavor because of their low odor threshold [33]. Results showed that the concentration of almost all alcohols were higher in the CHI/1% OA compared to the CON or other treatments (*p <* 0.05). Out of the alcohols, 3-methyl-1-butanol is produced by spoilage bacteria during refrigerated storage [49]. At 1 day, this compound was absent in all the samples, and it was only found in the CON after 21 days of storage. Pyrazines, as the Maillard reaction-derived products, contribute to the pleasant roasted aroma of cooked meat [33]. Out of them, two compounds (methyl pyrazine and 2,5-dimethyl pyrazine) were not found in the CON after 21 days of storage. This may be due to the fact that the bacterial spoilage-derived compounds associated with off-flavors limited the formation of these pyrazines. A hydrocarbon named butane was absent in all the samples at one day of storage, and it was only found in the CON. This compound might be produced by spoilage bacteria during storage. Taken together, it may be said that the coatings effectively protected the meat from the formation of volatile compounds associated with bacterial spoilage during storage, and the chitosan/oleic acid coating combination particularly increased the concentration of volatile aromas associated with pleasant odors, thus enhancing the flavor intensity of cooked meat.

## 4. Conclusions

This study, for the first time, assessed the applicability of chitosan/oleic acid edible coating in the preservation of fresh pork under aerobic packaging during storage. Coating with chitosan containing 1% oleic acid showed a stronger inhibitory effect against spoilage bacteria growth, TVBN formation and lipid oxidation in comparison with the CHI coating alone. After 21 days of storage, this coating completely inhibited the growth of *E. coli* and completely protected the meat from discoloration. In addition, it increased the concentration of aromatic volatiles associated with pleasant odors and inhibited the formation of volatiles derived from bacterial spoilage. Based on the guidelines of TVBN for freshness and APC for safety in countries, the incorporation of 0.5–1% oleic acid could extend the shelf life of fresh pork slices by up to 21 days under aerobic packaging conditions. This study suggests that the chitosan/oleic acid coating combination can be developed as a barrier technology in the preservation of fresh pork meat to extend shelf life and improve safety.

## Figures and Tables

**Figure 1 foods-11-01978-f001:**
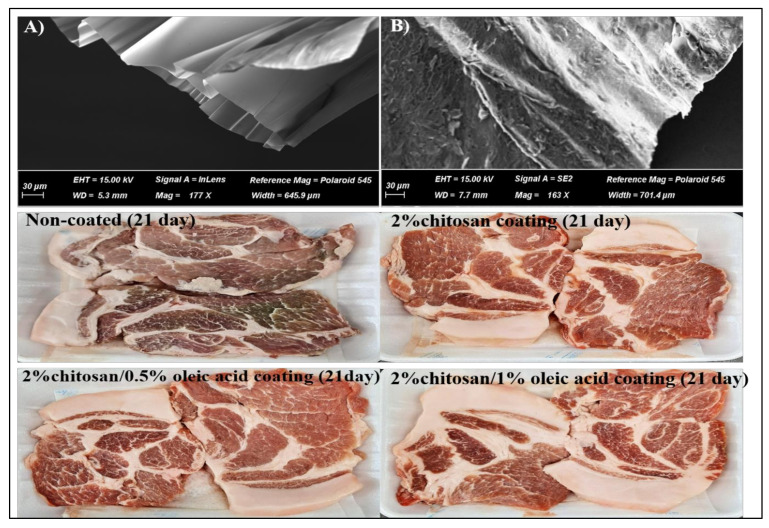
The representative images of the dry form of edible coatings: 2% chitosan (**A**), 1% oleic acid in 2% chitosan (**B**). The non-and coated pork slices wrapped with plastic film were taken on the 21st day of storage.

**Table 1 foods-11-01978-t001:** Effects of chitosan/oleic acid coating on fatty acid profiles of pork after one day of storage.

Items	CON	CHI	CHI/0.5% OA	CHI/1% OA
C14:0 (Myristic acid)	1.85 ± 0.01	1.70 ± 0.18	1.57 ± 0.05	1.52 ± 0.06
C16:0 (Palmitic acid)	27.45 ± 0.04	26.16 ± 0.80	26.13 ± 0.43	25.78 ± 0.75
C16:1n7 (Palmitoleic acid)	2.71 ± 0.02	2.06 ± 0.28	2.14 ± 0.45	2.09 ± 0.51
C18:0 (Stearic acid)	13.67 ± 0.75	14.42 ± 0.29	14.99 ± 0.35	13.95 ± 1.01
C18:1n9 (Oleic acid)	37.54 ± 0.74 ^c^	38.09 ± 1.43 ^c^	39.24 ± 0.28 ^ab^	40.66 ± 1.44 ^a^
C18:1n7 (Cis-vaccenic acid)	0.22 ± 0.03	0.24 ± 0.05	0.21 ± 0.03	0.23 ± 0.03
C18:2n6 (Linoleic acid)	14.79 ± 0.02	15.57 ± 0.14	14.06 ± 0.77	14.07 ± 0.59
C18:3n6 (Gamma linoleic acid)	0.01 ± 0.00	0.01 ± 0.00	0.01 ± 0.00	0.01 ± 0.00
C18:3n3 (Linolenic acid)	0.35 ± 0.03	0.36 ± 0.02	0.33 ± 0.03	0.34 ± 0.05
C20:1n9 (Eicosenoic acid)	0.90 ± 0.03	0.89 ± 0.01	0.89 ± 0.02	0.87 ± 0.03
C20:4n6 (Arachidonic acid)	0.35 ± 0.02	0.34 ± 0.06	0.32 ± 0.02	0.34 ± 0.04
C20:5n3 (Eicosapentaenoic acid)	0.01 ± 0.00	0.01 ± 0.00	ND	ND
C22:4n6 (Adrenic acid)	0.13 ± 0.02	0.13 ± 0.02	0.11 ± 0.01	0.13 ± 0.00
SFA	42.97 ± 0.76	42.29 ± 1.28	42.69 ± 0.55	41.25 ± 1.81
UFA	57.03 ± 0.76	57.71 ± 1.28	57.31 ± 0.55	58.75 ± 1.81
MUFA	41.37 ± 0.71 ^b^	41.28 ± 1.21 ^b^	42.47 ± 0.42 ^ab^	43.85 ± 1.16 ^a^
PUFA	15.65 ± 0.06 ^ab^	16.42 ± 0.09 ^a^	14.84 ± 0.82 ^b^	14.90 ± 0.68 ^b^
n6/n3	41.27 ± 3.53	44.09 ± 2.17	43.52 ± 2.38	43.56 ± 4.38
MUFA/SFA	0.96 ± 0.03 ^b^	0.98 ± 0.06 ^b^	1.00 ± 0.01 ^ab^	1.07 ± 0.07 ^a^
PUFA/SFA	0.36 ± 0.01	0.39 ± 0.01	0.35 ± 0.02	0.36 ± 0.03

CON, non-coating; CHI, 2% chitosan coating; CHI/0.5% OA, coating with 2% chitosan containing 0.5% (*v*/*v*) oleic acid; CHI/1% OA, coating with 2% chitosan containing 1% (*v*/*v*) oleic acid. SFA: Saturated fatty acid; UFA: Unsaturated fatty acid; MUFA: Monounsaturated fatty acid; PUFA: Polyunsaturated fatty acid. ND: Not detectable. Means with different superscripts ^(a–c)^ in a row are significantly different (*p* < 0.05).

**Table 2 foods-11-01978-t002:** Aerobic plate, *E. coli* and *pseudomonas* spp. counts of coated pork during storage.

Treatment	*Aerobic Plate Count (*log cfu/g*)*				*E. coli (*log cfu/g*)*				*Pseudomonas* spp. (log cfu/g)			
	1 Day	7 Day	14 Day	21 Day	1 Day	7 Day	14 Day	21 Day	1 Day	7 Day	14 Day	21 Day
CON	2.43 ± 0.17 ^d^	4.05 ± 0.03 ^c,A^	5.71 ± 0.13 ^b,A^	7.61 ± 0.01 ^a,A^	1.59 ± 0.26 ^b^	2.29 ± 0.03 ^a,A^	2.42 ± 0.07 ^a,A^	2.61 ± 0.03 ^aA^	1.97 ± 0.07 ^d^	3.96 ± 0.02 ^c,A^	4.57 ± 0.01 ^b,A^	5.58 ± 0.01 ^a,A^
CHI	2.35 ± 0.23 ^c^	3.65 ± 0.02 ^b,B^	4.04 ± 0.05 ^a,B^	4.11 ± 0.02 ^a,B^	1.62 ± 0.15 ^a,b^	1.82 ± 0.11 ^a,B^	1.72 ± 0.12 ^a,b,B^	1.36 ± 0.10 ^b,B^	1.88 ± 0.09 ^c^	3.52 ± 0.06 ^a,B^	3.41 ± 0.06 ^a,B^	2.25 ± 0.04 ^b,B^
CHI/0.5% OA	2.33 ± 0.10 ^c^	3.46 ± 0.04 ^b,C^	3.65 ± 0.02 ^a,b,C^	3.78 ± 0.01 ^a,C^	1.59 ± 0.11 ^a,b^	1.73 ± 0.05 ^a,C,B^	1.59 ± 0.05 ^b,C^	ND	1.84 ± 0.12 ^c^	3.47 ± 0.02 ^aC^	3.28 ± 0.02 ^b,C^	1.90 ± 0.05 ^c,C^
CHI/1% OA	2.31 ± 0.03 ^c^	3.38 ± 0.07 ^b,C^	3.55 ± 0.04 ^a,C^	3.62 ± 0.02 ^a,D^	1.53 ± 0.21	1.67 ± 0.06 ^C^	1.51 ± 0.19 ^C^	ND	1.88 ± 0.06 ^c^	3.40 ± 0.01 ^a,C^	3.14 ± 0.09 ^b,D^	1.52 ± 0.07 ^d,D^

CON, non-coating; CHI, 2% chitosan coating; CHI/0.5% OA, coating with 2% chitosan containing 0.5% (*v*/*v*) oleic acid; CHI/1% OA, coating with 2% chitosan containing 1% (*v*/*v*) oleic acid. Means with different superscripts ^(A–D)^ in a column are significantly different (*p* < 0.05). Means with different superscripts ^(a–d)^ in a row are significantly different (*p* < 0.05). ND: Not detectable.

**Table 3 foods-11-01978-t003:** TVBN and TBARS contents in coated pork during storage.

Treatment	TBARS (mg MAD/kg)				TVBN (mg/100 g)			
	1 Day	7 Day	14 Day	21 Day	1 Day	7 Day	14 Day	21 Day
CON	0.33 ± 0.01 ^d^	0.41 ± 0.04 ^c,A^	0.79 ± 1.20 ^b,A^	1.03 ± 0.04 ^a,A^	7.86 ± 0.77 ^d^	15.35 ± 0.94 ^c,A^	26.03 ± 1.86 ^b,A^	56.01 ± 2.40 ^a,A^
CHI	0.34 ± 0.01 ^d^	0.36 ± 0.02 ^c,B^	0.40 ± 0.04 ^B,b^	0.49 ± 0.03 ^a,B^	7.42 ± 0.98 ^d^	10.67 ± 0.31 ^c,B^	16.29 ± 0.69 ^b,B^	27.53 ± 1.84 ^a,B^
CHI/0.5% OA	0.32 ± 0.01 ^c^	0.36 ± 0.03 ^c,b,C^	0.39 ± 0.06 ^b,B,C^	0.46 ± 0.04 ^a,C^	7.36 ± 0.71 ^c^	9.73 ± 0.92 ^c,B^	15.73 ± 1.07 ^b,B^	19.79 ± 1.05 ^a,B,C^
CHI/1% OA	0.33 ± 0.01 ^c^	0.35 ± 0.02 ^c,B^	0.38 ± 0.03 ^b,C^	0.45 ± 0.04 ^a,C^	7.40 ± 0.62 ^c^	8.61 ± 0.68 ^c,B^	14.42 ± 0.55 ^b,B^	17.41 ± 1.85 ^a,C^

CON, non-coating; CHI, 2% chitosan coating; CHI/0.5%OA, coating with 2% chitosan containing 0.5% (*v*/*v*) oleic acid; CHI/1%OA, coating with 2% chitosan containing 1% (*v*/*v*) oleic acid. Means with different superscripts ^(A–C)^ in a column are significantly different (*p* < 0.05). Means with different superscripts ^(a–d)^ in a row are significantly different (*p* < 0.05).

**Table 4 foods-11-01978-t004:** Color traits, redness values reduction (%) and pH of coated pork during storage.

Treatment	L* (Lightness)				a* (Redness)			
	1 Day	7 Day	14 Day	21 Day	1 Day	7 Day	14 Day	21 Day
CON	52.15 ± 2.27 ^a^	51.81 ± 2.87 ^b^	50.29 ± 2.77 ^Bb^	48.29 ± 3.58 ^B,c^	17.35 ± 1.26 ^a^	15.12 ± 1.42 ^B,b^	13.85 ± 1.67 ^C,c^	10.65 ± 1.42 ^C,d^
CHI	50.88 ± 3.43 ^b^	53.72 ± 3.14 ^a^	51.80 ± 1.88 ^Aab^	52.50 ± 3.80A ^a,b^	16.74 ± 1.46 ^a^	16.11 ± 1.44 ^A,B,a^	15.05 ± 1.57 ^B,b^	14.12 ± 1.02 ^B,b^
CHI/0.5% OA	50.88 ± 2.29	52.73 ± 3.60	53.19 ± 3.23 ^A^	53.31 ± 2.98 ^A^	17.13 ± 1.62 ^a^	16.45 ± 1.60 ^A,a^	16.05 ± 1.11 ^A,B,a^	15.78 ± 1.33 ^A,B,b^
CHI/1% OA	52.81 ± 2.89	53.83 ± 2.00	51.98 ± 3.07 ^A^	52.24 ± 3.71 ^A^	17.16 ± 1.81	17.10 ± 1.36 ^A^	16.69 ± 1.18 ^A^	16.19 ± 2.01 ^A^
	Reduction percentage (%) of a* (redness) values				pH			
CON	0	12.87 ± 1.94 ^c,A^	20.14 ± 1.76 ^b,A^	38.59 ± 2.25 ^a,A^	6.03 ± 0.13 ^c^	6.10 ± 0.09 ^b^	6.28 ± 0.04 ^a,A^	6.33 ± 0.11 ^a,A^
CHI	0	3.76 ± 0.91 ^c,B^	10.04 ± 022 ^b,B^	15.61 ± 1.99 ^a,B^	6.02 ± 0.10 ^b^	6.03 ± 0.09 ^a,b^	6.06 ± 0.06 ^a,b,B^	6.12 ± 0.33 ^a,B^
CHI/0.5% OA	0	3.99 ± 0.98 ^b,B^	6.28 ± 0.58 ^a,B,C^	7.90 ± 0.16 ^a,B,C^	6.02 ± 0.10	6.03 ± 0.09	6.09 ± 0.06 ^C,B^	6.09 ± 0.06 ^C^
CHI/1% OA	0	0.36 ± 0.09 ^B^	2.71 ± 0.16 ^C^	5.63 ± 0.62 ^C^	6.02 ± 0.14	6.03 ± 0.06	6.03 ± 0.02 ^C^	6.06 ± 0.08 ^C^

CON, non-coating; CHI, 2% chitosan coating; CHI/0.5%OA, coating with 2% chitosan containing 0.5% (*v*/*v*) oleic acid; CHI/1%OA, coating with 2% chitosan containing 1% (*v*/*v*) oleic acid. Means with different superscripts ^(A–C)^ in a column are significantly different (*p* < 0.05). Means with different superscripts ^(a–d)^ in a row are significantly different (*p* < 0.05).

**Table 5 foods-11-01978-t005:** Amount (µg/g) of volatile aroma compounds in coated pork during storage.

Items	Retention Time (min)	Sotage Day	CON	CHI	CHI/0.5% OA	CHI/1% OA	Identying Method
Aldehydes							
2-Methyl pentanal	1.660	1	0.008 ± 0.000	0.012 ± 0.001	0.012 ± 0.000	0.019 ± 0.000	MS, STD
		21	NF	NF	NF	NF	
2-Methyl propanal	1.824	1	0.001 ± 0.000 ^b^	0.004 ± 0.000 ^a^	0.002 ± 0.000 ^a,b^	0.002 ± 0.000 ^a,b^	MS, STD
		21	0.001 ± 0.000	0.005 ± 0.000	0.002 ± 0.000	0.003 ± 0.000	
Butanal	1.995	1	NF	NF	NF	NF	MS, STD
		21	0.018 ± 0.000	0.008 ± 0.000	NF	NF	
2-Ethylhexanal	2.023	1	NF	NF	NF	NF	MS, STD
		21	0.033 ± 0.004	NF	NF	NF	
3-Methyl butanal	2.501	1	0.002 ± 0.000	0.036 ± 0.000	0.022 ± 0.008 ^B^	0.019 ± 0.001	MS, STD
		21	0.030 ± 0.001	0.014 ± 0.003	0.040 ± 0.005 ^A^	0.007 ± 0.000	
2-Methyl butanal	2.599	1	0.005 ± 0.000	0.008 ± 0.000	0.003 ± 0.000	0.005 ± 0.000	MS, STD
		21	0.018 ± 0.006	0.009 ± 0.000	0.013 ± 0.004	0.009 ± 0.000	
Pentanal	3.035	1	0.223 ± 0.034 ^B^	0.227 ± 0.073	0.193 ± 0.001 ^B^	0.260 ± 0.042	MS, STD
		21	0.085 ± 0.004 ^b,A^	0.314 ± 0.000 ^a,b^	0.461 ± 0.003 ^a,A^	0.349 ± 0.001 ^a,b^	
Hexanal	5.843	1	1.681 ± 0.064 ^b,A^	1.250 ± 0.138 ^b^	2.725 ± 0.006 ^a,b,B^	3.250 ± 0.408 ^a^	MS, STD
		21	0.841 ± 0.193 ^c,B^	1.948 ± 0.648 ^b^	3.543 ± 0.836 ^a,A^	3.952 ± 0.775 ^a^	
E,2-Hexenal	7.354	1	0.007 ± 0.000	0.004 ± 0.000	0.008 ± 0.000B	0.013 ± 0.008	MS, STD
		21	0.001 ± 0.000	0.011 ± 0.00	0.017 ± 0.00A	0.014 ± 0.00	
Heptanal	8.799	1	0.161 ± 0.039 ^b,A^	0.142 ± 0.049 ^b^	0.196 ± 0.022 ^b,B^	0.315 ± 0.073 ^a^	MS, STD
		21	0.033 ± 0.001 ^c,B^	0.261 ± 0.005 ^b^	0.378 ± 0.049 ^a,b,A^	0.453 ± 0.079 ^a^	
E,E-2,4-Heptadienal	9.526	1	NF	NF	0.002 ± 0.000 ^b^	0.009 ± 0.000 ^a^	MS, STD
		21	NF	NF	NF	NF	
E,2-Heptenal	10.279	1	0.027 ± 0.009 ^a,b,B^	0.022 ± 0.013 ^b^	0.036 ± 0.007 ^a,b^	0.059 ± 0.029 ^a^	MS, STD
		21	0.056 ± 0.001 ^A^	0.043 ± 0.002	0.076 ± 0.003	0.073 ± 0.006	
Benzaldehyde	10.349	1	0.043 ± 0.001	0.013 ± 0.001	0.014 ± 0.006	0.024 ± 0.005	MS, STD
		21	0.006 ± 0.000 ^c^	0.016 ± 0.003 ^b^	0.034 ± 0.007 ^a^	0.034 ± 0.002 ^a^	
Octanal	11.449	1	0.146 ± 0.016 ^c,A^	0.151 ± 0.015 ^b,c^	0.162 ± 0.009 ^b,B^	0.275 ± 0.088 ^a,B^	MS, STD
		21	0.081 ± 0.005 ^d,B^	0.179 ± 0.005 ^c^	0.281 ± 0.060 ^b,A^	0.355 ± 0.009 ^a,A^	
Benzeneacetaldehyde	12.391	1	0.003 ± 0.002	0.005 ± 0.002	0.002 ± 0.000	0.008 ± 0.000	MS, STD
		21	0.007 ± 0.000	0.011 ± 0.002	0.022 ± 0.005	0.011 ± 0.005	
Nonanal		1	0.076 ± 0.018 ^c^	0.072 ± 0.015 ^c^	0.120 ± 0.021 ^b^	0.179 ± 0.065 ^a,B^	MS, STD
		21	0.102 ± 0.009 ^b^	0.110 ± 0.035 ^b^	0.204 ± 0.106 ^b^	0.264 ± 0.017 ^a,A^	
E,2-Octenal	12.708	1	0.017 ± 0.005	0.013 ± 0.005	0.026 ± 0.007	0.033 ± 0.008	MS, STD
		21	0.008 ± 0.007	0.020 ± 0.008	0.032 ± 0.002	0.047 ± 0.008	
E,2-Nonenal	14.826	1	0.05 ± 0.000	0.003 ± 0.000	0.005 ± 0.000	0.009 ± 0.000	MS, STD
		21	0.001 ± 0.000	0.006 ± 0.000	0.014 ± 0.002	0.015 ± 0.000	
Decanal	15.711	1	0.008 ± 0.000	0.002 ± 0.000	0.006 ± 0.000	0.004 ± 0.000	MS, STD
		21	0.001 ± 0.000	0.008 ± 0.000	0.003 ± 0.000	0.003 ± 0.000	
E,E,2,4-Nonadienal	15.87	1	0.001 ± 0.000	0.001 ± 0.001	0.001 ± 0.000	0.011 ± 0.016	MS, STD
		21	0.001 ± 0.000	0.001 ± 0.000	0.003 ± 0.000	0.003 ± 0.000	
E,2-Decenal	16.745	1	0.004 ± 0.000	0.004 ± 0.000	0.003 ± 0.000	0.006 ± 0.000	MS, STD
		21	0.001 ± 0.000	0.004 ± 0.000	0.010 ± 0.003	0.009 ± 0.000	
E,E,2,4-Decadienal	17.321	1	0.001 ± 0.000	0.001 ± 0.000	0.005 ± 0.005	0.016 ± 0.011	MS, STD
		21	0.004 ± 0.000	0.001 ± 0.000	0.004 ± 0.000	0.003 ± 0.000	
2-Undecenal	18.519	1	0.002 ± 0.000	0.002 ± 0.000	0.001 ± 0.000	0.003 ± 0.000	MS, STD
		21	0.001 ± 0.000	0.002 ± 0.000	0.005 ± 0.000	0.005 ± 0.000	
Alcohols							
2-propanol	1.532	1	NF	NF	NF	NF	MS
		21	0.032 ± 0.000	NF	NF	NF	
2-Propen-1-ol	2.823	1	0.003 ± 0.000 ^B^	0.003 ± 0.000	0.003 ± 0.000	0.004 ± 0.000	MS
		21	0.01 ± 0.001 ^A^	0.007 ± 0.000	0.010 ± 0.000	0.008 ± 0.000	
1-Pentanol	4.619	1	0.191 ± 0.004	0.165 ± 0.008	0.167 ± 0.008 ^B^	0.246 ± 0.001 ^B^	MS, STD
		21	0.048 ± 0.002 ^c^	0.245 ± 0.021 ^b^	0.324 ± 0.003 ^a,b,A^	0.413 ± 0.001 ^a,A^	
3-Methyl-1-butanol	3.746	1	NF	NF	NF	NF	MS
		21	0.019 ± 0.008	NF	NF	NF	
1-Hexanol	7.879	1	0.025 ± 0.004 ^b^	0.025 ± 0.005 ^b^	0.029 ± 0.000 ^b^	0.045 ± 0.005 ^a^	MS, STD
		21	0.040 ± 0.021	0.116 ± 0.090	0.139 ± 0.005	0.453 ± 0.006	
1-Heptanol	10.655	1	0.021 ± 0.007 ^b,B^	0.017 ± 0.006 ^b^	0.032 ± 0.002 ^b^	0.056 ± 0.018 ^a^	MS, STD
		21	0.04 ± 0.000 ^A^	0.030 ± 0.003	0.048 ± 0.004	0.084 ± 0.006	
1-Octen-3-ol	10.885	1	0.060 ± 0.000 ^B^	0.037 ± 0.002 ^B^	0.005 ± 0.000 ^B^	0.011 ± 0.002 ^B^	MS, STD
		21	0.18 ± 0.002 ^a,b,A^	0.142 ± 0.053 ^b,A^	0.163 ± 0.027 ^b,A^	0.259 ± 0.065 ^a,A^	
E,2-Octen-1-ol	12.929	1	0.005 ± 0.000	0.004 ± 0.000	0.012 ± 0.004	0.015 ± 0.007	MS
		21	0.003 ± 0.000	0.009 ± 0.000	0.011 ± 0.000	0.038 ± 0.005	
1-Octanol	12.989	1	0.012 ± 0.004 ^b^	0.010 ± 0.002 ^b^	0.017 ± 0.005 ^b^	0.027 ± 0.005 ^a^	MS, STD
		21	0.017 ± 0.005	0.015 ± 0.007	0.024 ± 0.008	0.041 ± 0.000	
Ketones							
2-Butanone	2.022	1	0.004 ± 0.000	0.006 ± 0.000	0.005 ± 0.000	0.005 ± 0.000	MS, STD
		21	0.005 ± 0.000	0.005 ± 0.000	0.004 ± 0.000	0.005 ± 0.000	
Heptanone	8.394	1	0.05 ± 0.006	0.01 ± 0.000	0.021 ± 0.000 ^B^	0.02 ± 0.000 ^B^	MS, STD
		21	0.033 ± 0.002 ^b^	0.026 ± 0.002 ^b^	0.054 ± 0.005 ^a,b,A^	0.070 ± 0.000 ^aA^	
2,3-Butanedione	1.972	1	NF	NF	NF	NF	MS, STD
		21	0.020 ± 0.005	NF	NF	NF	
Pyrazines and sulfur-containing compounds							
Methylpyrazine	6.383	1	0.001 ± 0.000 ^b^	0.001 ± 0.000 ^b^	0.004 ± 0.000 ^b^	0.022 ± 0.009 ^a^	MS, STD
		21	NF	0.002 ± 0.00	0.003 ± 0.000	0.031 ± 0.006	
2,5-Dimethylpyrazine	9.014	1	0.010 ± 0.009	0.020 ± 0.008	0.008 ± 0.000	0.025 ± 0.009	MS, STD
		21	NF	0.008 ± 0.004	0.007 ± 0.000	0.035 ± 0.007	
3-Ethyl-2,5-dimethylpyrazine	13.172	1	0.001 ± 0.000 ^b^	0.005 ± 0.000 ^a,b^	0.004 ± 0.000 ^a,b^	0.008 ± 0.000 ^a^	MS
		21	0.06 ± 0.000	0.05 ± 0.000	0.06 ± 0.000	0.09 ± 0.000	
Methanethiol	1.486	1	0.002 ± 0.000	0.004 ± 0.000	0.005 ± 0.000	0.003 ± 0.000	MS
		21	0.002 ± 0.000	0.004 ± 0.000	0.003 ± 0.000	0.002 ± 0.000	
Carbon disulfide	1.754	1	0.002 ± 0.000	0.006 ± 0.000	0.004 ± 0.000	0.005 ± 0.000	MS, STD
		21	0.002 ± 0.000	0.005 ± 0.000	0.026 ± 0.034	0.006 ± 0.000	
Hydrocarbons							
1,2-Propanediol	1.523	1	0.072 ± 0.001 ^b^	0.254 ± 0.161 ^a^	0.116 ± 0.008 ^a,b^	0.135 ± 0.001 ^a,b^	MS
		21	0.004 ± 0.000 ^b^	0.083 ± 0.001 ^b^	0.035 ± 0.001 ^b^	0.291 ± 0.004 ^a^	
Butane	1.609	1	NF	NF	NF	NF	MS
		21	0.031 ± 0.013	NF	NF	NF	
2-Octene	5.318	1	0.003 ± 0.000	0.004 ± 0.000	0.001 ± 0.000	0.004 ± 0.000	MS
		21	0.004 ± 0.000	0.003 ± 0.000	0.008 ± 0.000	0.005 ± 0.000	
Xylene	7.803	1	0.003 ± 0.000 ^B^	0.004 ± 0.000	0.004 ± 0.000	0.008 ± 0.000 ^B^	MS
		21	0.024 ± 0.001 ^A^	0.011 ± 0.005	0.028 ± 0.008	0.016 ± 0.000 ^A^	
Ethylbenzene	7.553	1	0.001 ± 0.000	0.002 ± 0.000	0.002 ± 0.000 ^B^	0.003 ± 0.000	MS, STD
		21	0.016 ± 0.000	0.004 ± 0.000	0.009 ± 0.000 ^A^	0.006 ± 0.000	
Styrene	8.439	1	NF	NF	NF	NF	MS, STD
		21	0.010 ± 0.000 ^a^	0.003 ± 0.000 ^b^	NF	NF	
Heptanoic acid	11.149	1	0.010 ± 0.000	0.028 ± 0.008 ^B^	0.064 ± 0.006	0.160 ± 0.005	MS, STD
		21	0.014 ± 0.000	0.290 ± 0.008 ^A^	0.345 ± 0.009	0.249 ± 0.005	
Octanoic acid	15.079	1	0.004 ± 0.000 ^B^	0.008 ± 0.000	0.009 ± 0.000	0.004 ± 0.000	MS, STD
		21	0.010 ± 0.000 ^A^	0.030 ± 0.000	0.026 ± 0.002	0.005 ± 0.000	
Dodecane	15.58	1	0.017 ± 0.001	0.049 ± 0.000	0.050 ± 0.001	0.064 ± 0.009	MS, STD
		21	0.004 ± 0.000	0.006 ± 0.000	0.004 ± 0.000	0.067 ± 0.001	

MS: The compounds were identified by mass spectra from library; STD: The compounds were identified by external standards. CON, non-coating; CHI, 2% chitosan coating; CHI/0.5% OA, coating with 2% chitosan containing 0.5% (*v*/*v*) oleic acid; CHI/1%OA, coating with 2% chitosan containing 1% (*v*/*v*) oleic acid. Means with different superscripts ^(A,B)^ in a column are significantly different (*p* < 0.05). Means with different superscripts ^(a–c)^ in a row are significantly different (*p* < 0.05). NF: Not found.

## Data Availability

The data presented in this study are available on request from the corresponding author.

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
