# Peer review of "Application of a Newly Developed Chitosan/Oleic Acid Edible Coating for Extending Shelf-Life of Fresh Pork"

_foods, 2022, doi:10.3390/foods11131978_

Round 1
Reviewer 1 Report
The paper by Hoa et al. aimed to study the effects of the application of chitosan/oleic acid edible coating combination on the shelf-life of fresh pork aerobically packaged during storage.
The specific comments are as follows:
1. Introduction
L56: Consider to change “…development of of cooked…” to “…development of cooked…”.
2. Materials and Methods
L67: Indicate the reference number (28).
L72: Consider to change ”Instrument, Zeiss Co., Germany, Japan…” to ”Instrument, Zeiss Co., Germany…”.
L79: Consider to change ”meat samples were served as a control…” to ”meat samples were used as control…”.
L79. Authors should indicate that the first sampling moment was the day 1 of the study.
L89: Consider to change “…(log10 cfu/g).”to “…(log cfu/g).”, throughout the manuscript.
L93: Please clarify: how many different locations of muscle tissues were used?
L96: Authors should briefly describe the method used.
L111: Consider to change “…The extracted lipid…” to “Briefly, the extracted lipid…”.
L129: Consider to change “…For each the storage period…” to “For each storage period…”.
3. Results and Discussion
L160: Consider to change “…Means in a same column with different superscripts (A,B,C)…” to “Means in a same column with different superscripts (A,B,C, D)…”.
L180: Correct the reference “Fang et al.” number.
Table 3: Superscripts meaning for significance level are missing.
L240: Consider to change “…Consistent with our results, numerous other studies…” to “…Consistent with our results, other studies…”.
L263: Consider to change “…with 23 compounds were identified…” to “…with 23 compounds identified…”.
L274: Consider to change “…as the Mallard reaction-derived…” to “…as the Maillard reaction-derived…”.
The n value is missing in all tables.
Author Response
Thank you very much for your efforts in improving the quality of our manuscript.

Reviewer 2 Report
The work is interesting and shows the use of edible coatings in order to extend the durability of pork.
Please add "pork" or "fresh pork" to your keywords.
To introduction section please add a few sentences about other edible coatings used for meat
Line 47: "Chitosan has an potent" - a potent
Line 57: double of
Line 73: What muscle was used for the research?
Line 108: Where was the sample taken for testing fatty acid composition? Whether it included an edible coating?
Line 133: Maybe higher level of oleic acid could be due to the coating present in the sample?
Author Response

(The authors gave the same response as above.)

Reviewer 3 Report
The manuscript is focused on the use of a chitosan/oleic acid edible coating to extend the Shelf-life of fresh pork. The topic is of interest to the field of food packaging.
- Throughout the paper the Authors use improperly the term shelf life. To assess the shelf life of a given packaged food product first the quality indices of the product along with their threshold value must be identified. Each quality indices must be monitored during storage, and the storage time at which it reaches its threshold valued must be determined. The packaged product shelf life is defined as the shortest storage time at which a quality index reaches its threshold value. The authors are invited to emend the paper accordingly.
- The authors should explain the reason why they have focused their attention on three microbial group (i.e., Aerobic Plate Count (APC), E. coli, Pseudomonas spp.). The Authors should also indicate at least 1 bibliographic reference that support their choice.
- The Authors should indicate at which storage time the data listed in Table 1 refer.
Author Response

(The authors gave the same response as above.)
